# Moving Multitarget Detection Using a Multisite Radar System with Widely Separated Stations

Shiyu Zhang [1], Yu Zhou [1,*], Minghui Sha [2], Linrang Zhang [1] and Lan Du [1]

[1] National Laboratory of Radar Signal Processing, Xidian University, Xi'an 710071, China; zhangshiyu@stu.xidian.edu.cn (S.Z.); lrzhang@xidian.edu.cn (L.Z.); dulan@mail.xidian.edu.cn (L.D.)
[2] Beijing Institute of Radio Measurement, Beijing 100854, China; shaminghui1234@163.com
[*] Correspondence: zhouyu@mail.xidian.edu.cn

**Abstract:** This study investigates the detection problem of multiple moving targets using a multisite radar system with widely separated stations. Spatial mapping is presented to integrate the observation data of a moving target from different angles into a spatial resolution cell (SRC). However, data association errors occur in some SRCs in this way, which causes extra false alarm, called the "ghost target". Therefore, an interference discriminator-based detector is developed. In this way, the background interference is discriminated between "ghost target" and pure noise, and then the final decision is made based on the generalized likelihood ratio test. Statistical analyses are provided to discuss the achievable performance. Simulation results show that the proposed algorithm can accurately detect multiple moving targets while suppressing the "ghost target".

**Keywords:** multisite radar system; multitarget detection; generalized likelihood ratio testing

## 1. Introduction

A multisite radar system (MSRS) is a comprehensive system wherein signals from all separated stations are fused and jointly processed [1]. In an MSRS, information is extracted from several spatially separated regions of scattered (or radiated by signal sources) fields, which allows improved target detection [2–4], target localization [5–7], and interference proofing [8–10]. The MSRS with widely separated stations captures diversity gain and geometry gain because of observations from different angles. This attribute enhances the target detection performance [11].

The detection problem has been the focus of continuous research for several decades. Researchers have proposed many adaptive detectors, such as the subspace-based detector [12], the Rao-base detector [13], and the information geometry-based detector [14]. The MSRS detection methods can be regarded as the extensions of these adaptive detectors. Depending on the fusion level, the detection methods can be divided into two categories: distributed and centralized [15]. In the distributed method, each radar station uses its observation data to build a local statistic or make a local decision and then fuses the statistic or decision in the fusion center for decision-making. The fusion rule can be counting rule [16], weighted decision fusion [17], or generalized likelihood ratio test (GLRT) [18]. This approach is usually accompanied by performance loss because the local statistic lacks global information. However, MSRS complexity is reduced dramatically when distributed processing is adopted in the system [15,19]. The second category is centralized target detection. These methods make decisions about the presence of a target by comparing the global test statistic and a predetermined threshold in the signal fusion center. The noncoherent integration detector is first provided for the fluctuating signals [20,21]. Considering that the statistical characteristics of noise are unknown, several detection algorithms are proposed based on GLRT algorithm for different background noise [1,22–24]. The above algorithms focus on testing the measurement for a single target and assume that the time of arrival is a priori known.

Actually, the location of the target is unknown, so it is difficult to implement the registration that is the prerequisite for signal fusion [25]. Because of this situation, the signals received by different radars are integrated in a single spatial field [26]. Observation data of a target can be gathered in the spatial resolution cells (SRCs), which are defined as the intersection of the overlapping range bins (RBs). Thus, the detection problem is formulated as the statistical decision of each SRC within the common surveillance region [27,28]. Actually, integrating signals from different radars based on SRCs may cause an error in data association. The pure noise data may be associated with the observation data of the target in some SRCs that contain no targets. These SRCs can be detected with a certain probability, which leads to extra false alarms called the "ghost target" problem [27,29,30]. To detect an individual target, the peak search is used to find the SRC containing the target in [31]. However, the "ghost target" problem will be more severe in the presence of multiple targets. To solve this problem, a CLEAN-based multitarget detection method is proposed when the radar system observes motionless targets with only one pulse signal [32]. Radar systems are always exposed to complicated electromagnetic environments where clutter degrades the detection performance. Under this situation, moving target detection (MTD) methods using several pulses are required for clutter suppression [33]. In these methods, the differences in Doppler frequency have been exploited in the radar system to separate targets from clutter because clutter carries only small frequency shifts, whereas the target's movement causes larger frequency shifts. These existing MTD algorithms just focus on the detection of a single target and ignore the existence of "ghost targets" [34,35].

To the best of our knowledge, no work is conducted on moving multitarget detection for MSRS. In this study, we investigate the problem of MTD using MSRS with widely separated stations when multiple targets exist in the common surveillance region. We work on the MTD to detect the physical targets and reject the "ghost targets". Contributions in this paper are summarized as follows.

1.  A scheme of space mapping is presented to integrate the observation data of a moving target from different views. The observation data are gathered in the corresponding SRC, so the multitarget detection problem is split into several statistical decision problems in each SRC. The relationships between physical targets and "ghost targets" are analyzed.
2.  The interference discrimination (ID)-based detector is developed to overcome the "ghost target" problem. The detection problem is formulated based on ternary hypothesis testing and can be solved in two steps. The background interference is discriminated between "ghost target" and pure noise, and then the presence of target is decided based on GLRT.
3.  Experiments performed on simulation database verify the advantages of the proposed ID-based detector compared with the traditional GLRT detector.

The framework of this paper is designed as follows. In Section 2, the signal model of an MSRS is introduced. The detection method of multiple moving targets is developed in Section 3. The numerical simulation results are provided in Section 4. Section 5 shows the conclusion.

## 2. Signal Model

### 2.1. Received Signal

Consider an MSRS in a two-dimensional (range and azimuth) coordinate system. The MSRS consists of $I$ ($I > 1$) monostatic radars that are stationarily located at $\mathbf{p}_{\mathrm{R},i} \in \mathbb{R}^{2\times1}$, where $i = 0, 1, \ldots, I - 1$. Their antennas are separated widely enough to provide space diversity gain and geometry gain. A radar can only receive the signal transmitted itself, which can be achieved by designing orthogonal waveforms [36], or by simply using different frequency bands [37]. All radar sites already have time synchronization with a range cell level [38]. Each radar has only one signal channel and does not have angle-of-arrival detection capability [39]. The radar system cooperatively and synchronously observes a common surveillance region in which $J$ ($J > 0$) moving targets are located at

$\mathbf{p}_{\mathrm{T},j} \in \mathbb{R}^{2\times 1}$ and move with the velocities $\mathbf{v}_{\mathrm{T},j} \in \mathbb{R}^{2\times 1}$, where $j = 0, 1, \dots, J - 1$. Actually, $J$, $\mathbf{p}_{\mathrm{T},j}$, and $\mathbf{v}_{\mathrm{T},j}$ are unknown variables beforehand.

The Doppler frequency caused by the movement between the $j$th target and $i$th radar can be expressed as

$$f_{\mathrm{d},i,j} = \frac{2\left(\mathbf{p}_{\mathrm{R},i} - \mathbf{p}_{\mathrm{T},j}\right)^{\mathrm{T}} \mathbf{v}_{\mathrm{T},j}}{\lambda_i \left\| \mathbf{p}_{\mathrm{R},i} - \mathbf{p}_{\mathrm{T},j} \right\|}, \tag{1}$$

where $\|\cdot\|$ is the operator of 2-norm. $\lambda_i$ represents the wave length of the $i$th radar. Given that every radar transmits $M$ pulse signals with the pulse repetition interval (PRI) $T_{\mathrm{p}}$, the Doppler vector is obtained as

$$\mathbf{d}_i\left(f_{\mathrm{d},i,j}\right) = \left[1, e^{j2\pi f_{d,i,j} T_{\mathrm{P}}}, \dots, e^{j2\pi f_{d,i,j}\left((M-1)T_{\mathrm{P}}\right)}\right]^{\mathrm{T}}. \tag{2}$$

Assuming that all transmitted signals are observed irrespective of range migration, the baseband echo received by the $i$th radar is expressed as

$$\mathbf{R}_i = \sum_{j=0}^{J-1} \alpha_{i,j}\mathbf{d}_i^\dagger\left(f_{\mathrm{d},i,j}\right)\otimes\mathbf{s}_i(\tau_{i,j}) + \mathbf{W}_i, \tag{3}$$

$$\mathbf{s}_i(\tau_{i,j}) \triangleq \begin{bmatrix} s_i(0 - \tau_{i,j}) \\ s_i(T_s - \tau_{i,j}) \\ \vdots \\ s_i((N-1)T_s - \tau_{i,j}) \end{bmatrix}, \tag{4}$$

where $\mathbf{R}_i \in \mathbb{C}^{M\times N}$ represents the matrix form of the receiving signal, and the pulse number dimension is the slow time axis. $\otimes$ is the Kronecker product. $(\cdot)^\dagger$ denotes the conjugate transpose operation. $\alpha_{i,j}$ denotes the complex amplitude, which is a function of the transmitting power, propagation loss, radar cross section, and position of the target. $s_i(t)$ denotes the baseband transmitted waveforms of the $i$th radar, which is designed to facilitate the acquisition of channel separation. $N$ is the length of the signal, where $0 < N \leq \lfloor T_p/T_s \rfloor$. $T_s$ denotes the sample interval. The time delay from the $i$th radar reflected by the $j$th target is given as

$$\tau_{i,j} = \frac{2}{c}\sqrt{\left(\mathbf{p}_{\mathrm{R},i} - \mathbf{p}_{\mathrm{T},j}\right)^{\mathrm{T}}\left(\mathbf{p}_{\mathrm{R},i} - \mathbf{p}_{\mathrm{T},j}\right)}. \tag{5}$$

Obviously, $\tau_{i,j}$ is a function of $\mathbf{p}_{T,j}$, so the term $\mathbf{s}_i(\tau_{i,j})$ can be expressed as $\mathbf{s}_i(\mathbf{p}_{T,j})$. $\mathbf{W}_i \triangleq [\mathbf{w}_{i,0}, \mathbf{w}_{i,1}, \dots, \mathbf{w}_{i,M-1}]$ is the background interference that consists of the noise and clutter. $\mathbf{w}_{i,m}$ denotes the interference during the $m$th pulse. The distinct elements of the vector $\mathbf{w}_{i,m}$ are modeled as Gaussian random variables. Because of the widespread antennas and different carrier frequencies, the background noise is assumed to be independent between different radar sites

$$E\left\{\mathbf{w}_{i,m}\mathbf{w}_{i',m'}{}^\dagger\right\} = \begin{cases} \sigma_i^2\mathbf{I}_N & i = i' \text{ and } m = m' \\ \mathbf{0}_N & \text{others} \end{cases}, \tag{6}$$

where $\mathbf{I}_N$ represents the $N \times N$ identity matrix. $\sigma_i^2$ denotes the noise level of the $i$th radar. The sumption holds in the case that the surveillance region contains no highly correlated clutter, which is suitable for homogeneous environment such as air-search mode [32].

### 2.2. Spatial Mapping

The signals reflected by the same target are difficult to synthesize because observation data are collected in the local coordinate system and the location and velocity of all the

targets are uncertainties. To solve this problem, an SRC-based method is presented to integrate the signals from different angles.

Usually, the data from every radar are organized into an $N \times M$ matrix, where the column of samples denotes the slow-time signal, and the row of samples represents the fast-time signal. The slow-time signal can represent a series of measurements for an RB. Note that the range resolution is related to the bandwidth of the transmitted waveform, and the Nyquist rate in fast time is simply the bandwidth for a complex signal [33]. Thus, the width of RB can be expressed as $\Delta R = cT_s/2$ with the speed of light $c$. Figure 1 illustrates the relationship between the slow-time signals of the targets and the corresponding RBs. When the MSRS observes two targets for a coherent processing interval, the received sampling signals are stored in digital memory, where each cube denotes a single sample. The shaded row is the slow-time signal for the RB with the same color. The range bin corresponding to the slow-time signal of a target will intersect in a region where the observation data of a target can be integrated. Thus, target detection can be implemented in the spatial domain instead of the time domain by testing each SRC within the common surveillance region.

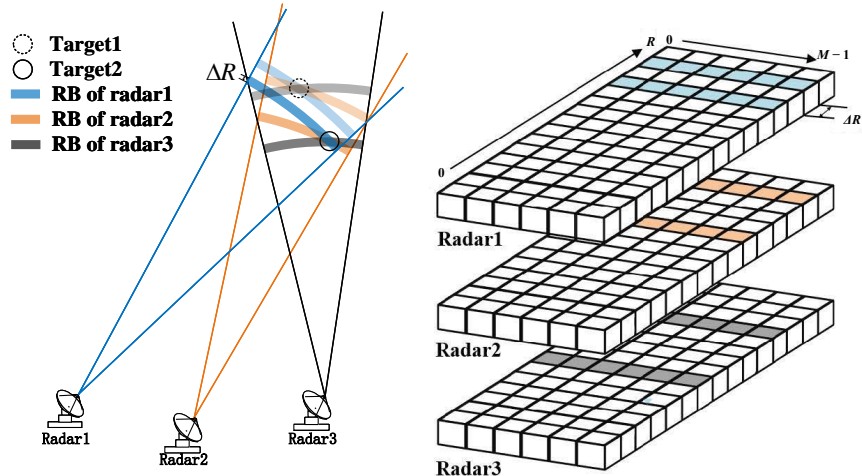

**Figure 1.** Relationship between range bins and the received signals.

To integrate signals received from multiple radars into a single air picture, each signal of the local radar must be expressed in a common coordinate system. Assume that $G$ SRCs, which are denoted as $c(\mathbf{p}_{C,g})$ with the center $\mathbf{p}_{C,g} = [x_{C,g}, y_{C,g}]^{\mathrm{T}}$, can cover the common surveillance area. Thus, the signal from different angle can be integrated into these SRCs in the common surveillance region. The range bin matched filter samples of $c(\mathbf{p}_{C,g})$ in $i$th radar are given as

$$
\mathbf{z}_i(\mathbf{p}_{C,g}) = \begin{cases} \mathbf{R}_i^{\dagger}\mathbf{s}_{i,g} & \left| \theta(\mathbf{p}_{C,g}) - \theta_0 \right| \leq \Delta\theta/2 \\ 0 & \left| \theta(\mathbf{p}_{C,g}) - \theta_0 \right| > \Delta\theta/2 \end{cases}, \tag{7}
$$

where $\mathbf{s}_{i,g} = \mathbf{s}_i(\mathbf{p}_{C,g})$ is the reference signal of the $g$th SRC, and $\theta(\mathbf{p}_{C,g})$ represents the angle of the $g$th SRC. $\theta_0$ denotes the beam direction. $\Delta\theta$ is the beamwidth. $\mathbf{z}_{i,g}$ is the slow time signal of $c(\mathbf{p}_{C,g})$ received by the $i$th radar. $\mathbf{Z}_g \triangleq [\mathbf{z}_{0,g}, \mathbf{z}_{1,g}, \dots, \mathbf{z}_{I-1,g}]$ is defined as the observation matrix of the $g$th SRC, which consists of slow-time signals from different radars. Because the signals are transfered in the surveillance space, (7) can be regarded as the space mapping from the local coordinate reference system of the common Cartesian coordinate system.

In this SRC-based method, the SRC is considered the minimum cell for signal fusion. These SRCs have a symbiotic relationship, which can cause a certain number of false alarms [31]. Some definitions are introduced to discuss the relationship. If a given SRC

$c(\mathbf{p}_{C,g})$ is covered by the beams of the MSRS, the SRC set corresponding to the same RB is expressed as

$$\mathbb{Q}_i(\mathbf{p}_{C,g}) = \left\{ c(\mathbf{p}) \Big| \big|\tau_i(\mathbf{p}_{C,g}) - \tau_i(\mathbf{p})\big| < cT_s/2, \mathbf{p} \in \mathbb{R}^{2\times1} \right\}. \tag{8}$$

The slow signal $\mathbf{z}_i(\mathbf{p}_{C,g})$ the SRCs belonging to $\mathbb{Q}_i(\mathbf{p}_{C,g})$. In addition, the family of $c(\mathbf{p}_{C,g})$ is defined as

$$\mathbb{F}(\mathbf{p}_{C,g}) = \bigcup_i^{I-1} \mathbb{Q}_i(\mathbf{p}_{C,g}). \tag{9}$$

The SRCs in $\mathbb{F}(\mathbf{p}_{C,g})$ shares the same RB with $\mathbf{p}_{C,g}$. An arbitrary SRC can belong to $I$ families simultaneously. The dominant SRC (DSRC) of the family is defined as

$$\mathbb{Y}(\mathbf{p}_{C,g}) = \bigcap_i^{I-1} \mathbb{Q}_i(\mathbf{p}_{C,g}). \tag{10}$$

The relative SRC (RSRC) subject to the DSRC is defined as

$$\mathbb{O}(\mathbf{p}_{C,g}) = \mathbb{F}(\mathbf{p}_{C,g}) - \mathbb{Y}(\mathbf{p}_{C,g}). \tag{11}$$

If there is a target in the DSRC, the RSRCs usually share more than one RB corresponding to the target [40]. So the RSRC have an extra false alarm rate, which is defined as the "ghost target".

## 3. Detection of Multiple Moving Targets

After spatial mapping, the detection problem of multiple moving targets can be divided into several decisions about the presence of each potential target. As shown in Figure 2, if an SRC contains a target, all members in the family are affected by the target. Total observation data can be gathered in the DSRC when only incomplete observation data of the target can be integrated in the RSRC.

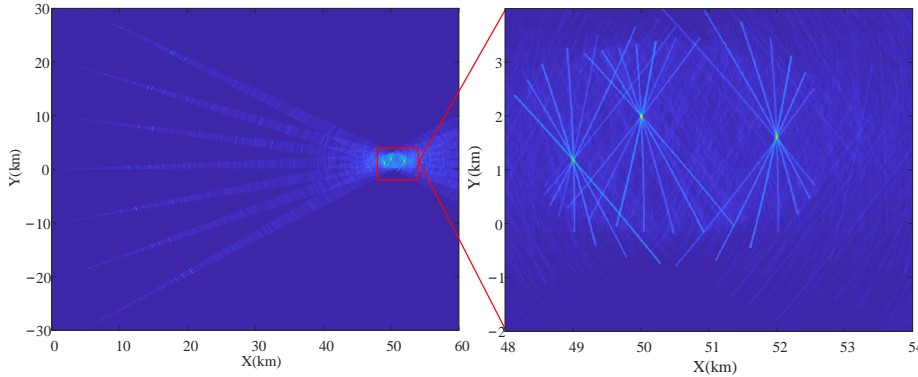

**Figure 2.** Energy accumulation in space.

We have to decide among more than two possible hypotheses under this situation. Actually, binary hypothesis testing is widely used in radar detection, noise only versus signal plus noise [18,41,42]. However, the RSRC of a target can be detected with a certain probability if the traditional binary hypothesis is taken to design the detector. Therefore, given an SRC $\mathbf{p}_{C,g}$, we formulate the detection problem as ternary hypothesis testing:

$$\begin{cases} \mathcal{H}_0\text{:}\mathbf{Z}_g = \mathbf{N}_g \\ \mathcal{H}_1\text{:}\mathbf{Z}_g = \mathbf{Y}_T + \mathbf{N}_g \,, \\ \mathcal{H}_2\text{:}\mathbf{Z}_g = \mathbf{Y}_R + \mathbf{N}_g \end{cases} \tag{12}$$

where $\mathbf{Y}_T \triangleq [\mathbf{y}_{0,g}, \mathbf{y}_{1,g}, \ldots, \mathbf{y}_{I-1,g}]$, $\mathbf{Y}_R \triangleq [0, \ldots, \mathbf{y}_{l,g}, \ldots, 0]$, $\mathbf{N}_g \triangleq [\mathbf{n}_{0,g}, \mathbf{n}_{1,g}, \ldots, \mathbf{n}_{I-1,g}]$, $\mathbf{y}_{i,g} = \alpha_{i,g} \mathbf{s}_{i,g}^\dagger \mathbf{s}_{i,g} \mathbf{d}_i(f_{\mathrm{d},i,g})$, and $\mathbf{n}_{i,g} \sim \mathcal{CN}(0, \sigma_i^2 \mathbf{s}_{i,g}^\dagger \mathbf{s}_{i,g} \mathbf{I}_N)$. $\mathcal{H}_0$ is the hypothesis of pure noise. The noise level $\sigma_i^2$ is assumed to be known. $\mathcal{H}_1$ is the present hypothesis, i.e., $\mathrm{c}(\mathbf{p}_{C,g}) \in \mathbb{Y}(\mathbf{p}_{T,j})$. Under $\mathcal{H}_2$, $\mathrm{c}(\mathbf{p}_{C,g}) \in \mathbb{F}(\mathbf{p}_{T,j})$, the observation data include a partial target component, because the SRC is influenced by the DSRC containing a target but contains no targets. We assume that the SRC shares the $l$th RB with an unknown target. Then, the conditional probability distribution functions (PDF) under $\mathcal{H}_0$, $\mathcal{H}_1$, and $\mathcal{H}_2$ can be expressed respectively as

$$f(\mathbf{Z}_g | \mathcal{H}_0) = \prod_{i=0}^{I-1} \frac{1}{\pi \mathbf{s}_{i,g}^\dagger \mathbf{s}_{i,g} \sigma_i} \exp\left\{ -\frac{1}{\mathbf{s}_{i,g}^\dagger \mathbf{s}_{i,g} \sigma_i} \mathbf{z}_{i,g}^\dagger \mathbf{z}_{i,g} \right\}, \tag{13}$$

$$\begin{aligned} f(\mathbf{Z}_g | \mathbf{a}_g, \mathbf{f}_{\mathrm{d},g}, \mathcal{H}_1) \\ = \prod_{i=0}^{I-1} \frac{1}{\pi \mathbf{s}_{i,g}^\dagger \mathbf{s}_{i,g} \sigma_i} \exp\left\{ -\frac{1}{\mathbf{s}_{i,g}^\dagger \mathbf{s}_{i,g} \sigma_i} \left( \mathbf{z}_{i,g} - \alpha_{i,g} \mathbf{s}_{i,g}^\dagger \mathbf{s}_{i,g} \mathbf{d}_i(f_{\mathrm{d},i,g}) \right)^\dagger \right. \\ \left. \times \left( \mathbf{z}_{i,g} - \alpha_{i,g} \mathbf{s}_{i,g}^\dagger \mathbf{s}_{i,g} \mathbf{d}_i(f_{\mathrm{d},i,g}) \right) \right\}, \end{aligned} \tag{14}$$

$$\begin{aligned} f(\mathbf{Z}_g | \alpha_{l,g}, f_{\mathrm{d},l,g}, \mathcal{H}_2) = \frac{1}{\pi \mathbf{s}_{l,g}^\dagger \mathbf{s}_{l,g} \sigma_l} \exp\left\{ -\frac{1}{\mathbf{s}_{l,g}^\dagger \mathbf{s}_{l,g} \sigma_l} \left( \mathbf{z}_{l,g} - \alpha_{l,g} \mathbf{s}_{l,g}^\dagger \mathbf{s}_{l,g} \mathbf{d}_l(f_{\mathrm{d},l,g}) \right)^\dagger \left( \mathbf{z}_{l,g} - \alpha_{l,g} \mathbf{s}_{l,g}^\dagger \mathbf{s}_{l,g} \mathbf{d}_l(f_{\mathrm{d},l,g}) \right) \right\} \\ \times \prod_{i=0,i\neq l}^{I-1} \frac{1}{\pi \mathbf{s}_{i,g}^\dagger \mathbf{s}_{i,g} \sigma_i} \exp\left\{ -\frac{1}{\mathbf{s}_{i,g}^\dagger \mathbf{s}_{i,g} \sigma_i} \mathbf{z}_{i,g}^\dagger \mathbf{z}_{i,g} \right\}, \end{aligned} \tag{15}$$

where $\mathbf{a}_g = [\alpha_{0,g}, \alpha_{2,g}, \ldots, \alpha_{I-1,g}]^\mathrm{T}$ denotes the amplitue vector of $g$th potential target. $\mathbf{f}_{\mathrm{d},g} = [f_{\mathrm{d},0,g}, f_{\mathrm{d},1,g}, \ldots, f_{\mathrm{d},I-1,g}]^\mathrm{T}$ is the unknown Doppler frequency.

### 3.1. Interference Discrimination-Based Detection Method

The goal of radar detection is to determine whether a target is present in an SRC. There are no requirements to decide if $\mathcal{H}_2$ is true. Under $\mathcal{H}_2$, the target component in the observation data can be considered interference from the DSRC. Consequently, the detection problem can be distinguishing $\mathcal{H}_1$ from $\mathcal{H}_0$ and $\mathcal{H}_2$:

$$L(\mathbf{Z}_g) \underset{\mathcal{H}_0 + \mathcal{H}_2}{\overset{\mathcal{H}_1}{\gtrless}} \xi, \tag{16}$$

where $L(\mathbf{Z}_g)$ is a statistic. $\xi$ is the threshold that is a function of the false alarm. A two-step method is proposed to solve the detection problem. First a discriminator is designed to decide the type of background interference, and then the final decision rules are derived according to the discrimination result.

#### 3.1.1. Interference Discrimination

For choice of decision rule, the discrimination of background interference is required. Under $\mathcal{H}_2$, the observation data consists of partial target data and pure noise. Under $\mathcal{H}_2$, the observation data from the target is quite different from other components. In fact the "ghost target" can exceed the threshold because it contains the target components, by which

we can distinguish the background interference. Thus the discrimination problem can be built as

$$
\begin{cases}
\text{Case I:} & \alpha_{l,g} = 0 \\
\text{Case II:} & \alpha_{l,g} \neq 0
\end{cases}
\tag{17}
$$

when $\alpha_{l,g} = 0$, the observation data contains no target components. Otherwise, it may be a "ghost target". For testing the the parameter $\alpha_{l,g}$, we build the likelihood ratio

$$
\tilde{D}(\mathbf{Z}_g) = \frac{f(\mathbf{Z}_g | \alpha_{l,g}, f_{\mathrm{d},l,g}, \mathcal{H}_2)}{f(\mathbf{Z}_g | \mathcal{H}_0)}
\tag{18}
$$

By simplifying (18), the discrimination statistic can be obtained

$$
\begin{aligned}
\ln \tilde{D}(\mathbf{Z}_g) &\propto -\left( \mathbf{z}_{l,g} - \alpha_{l,g} \mathbf{s}_{l,g}^\dagger \mathbf{s}_{l,g} \mathbf{d}\left(f_{\mathrm{d},l,g}\right) \right)^\dagger \left( \mathbf{z}_{l,g} - \alpha_{l,g} \mathbf{s}_{l,g}^\dagger \mathbf{s}_{l,g} \mathbf{d}\left(f_{\mathrm{d},l,g}\right) \right) + \mathbf{z}_{l,g}^\dagger \mathbf{z}_{l,g} \\
&= -2\Re\left\{ \alpha_{i,g}^* \mathbf{s}_{l,g}^\dagger \mathbf{s}_{l,g} \mathbf{d}^\dagger\left(f_{\mathrm{d},l,g}\right) \mathbf{z}_{l,g} \right\} + \left| \alpha_{l,g} \mathbf{s}_{l,g}^\dagger \mathbf{s}_{l,g} \mathbf{d}^\dagger\left(f_{\mathrm{d},l,g}\right) \right|^2.
\end{aligned}
\tag{19}
$$

where $\Re\{\cdot\}$ is the real part of a complex value. $\alpha_{l,g}$, $f_{\mathrm{d},l,g}$, and $l$ are unknown parameters that need to be etsimated. Maximizing (19) with respect to $\alpha_{l,g}$ yields

$$
\hat{\alpha}_{l,g} = \max_{l, f_{\mathrm{d},l,g}} \frac{\mathbf{d}_l^\dagger\left(f_{\mathrm{d},l,g}\right) \mathbf{z}_{l,g}}{\mathbf{s}_{l,g}^\dagger \mathbf{s}_{l,g} \mathbf{d}_l^\dagger(f_{\mathrm{d},l,g}) \mathbf{d}_l(f_{\mathrm{d},i,g})}.
\tag{20}
$$

The estimated amplitude of interference is equivalent to the largest elements in the observation matrix. Substituting (20) into (18), we have

$$
D(\mathbf{Z}_g) = \max_{l, f_{\mathrm{d},l,g}} \frac{\left| \mathbf{d}_l^\dagger\left(f_{\mathrm{d},l,g}\right) \mathbf{z}_{i,g} \right|^2}{\sigma_l^2 \mathbf{s}_{l,g}^\dagger \mathbf{s}_{l,g} \mathbf{d}_l^\dagger(f_{\mathrm{d},l,g}) \mathbf{d}_l(f_{\mathrm{d},l,g})},
\tag{21}
$$

where $\sigma_l^2$ can be estimated in practice:

$$
\hat{\sigma}_{l,g}^2 = \frac{1}{(I-1)} \sum_{i=0, i \neq l}^{I-1} \frac{\left| \mathbf{d}_i^\dagger\left(f_{\mathrm{d},i,g}\right) \mathbf{z}_{i,g} \right|^2}{\mathbf{s}_{l,g}^\dagger \mathbf{s}_{l,g} \mathbf{d}_l^\dagger(f_{\mathrm{d},l,g}) \mathbf{d}_l(f_{\mathrm{d},l,g})}.
\tag{22}
$$

By substituting (22), the discrimination rule can be obtained: decide case II if

$$
D(\mathbf{Z}_g, \kappa) = \max_{l, f_{\mathrm{d},l,g}} \left\{ \frac{(I-1)\left| \mathbf{d}_l^\dagger\left(f_{\mathrm{d},l,g}\right) \mathbf{z}_{l,g} \right|^2}{\sum_{i=0, i \neq l}^{I-1} \left| \mathbf{d}_i^\dagger\left(f_{\mathrm{d},i,g}\right) \mathbf{z}_{i,g} \right|^2} \right\} > \eta,
\tag{23}
$$

where $\eta$ is the threshold of the discriminator. The threshold is designed according to the principle of constant $P_{\mathrm{e}} = P\{\text{case I} | \text{case I}\}$. Because $\mathcal{H}_1$ is ignored in (21), a decision between the absent or present hypothesis is required. The decision rules will be developed according to the intereference type.

### 3.1.2. Target Detection Method

As shown in Figure 3, the decision rule is selected according to the interference type. Depending on the type of background interference, we have two cases to solve the above detection problem. Case I: if the interference background is pure noise, the decision between $\mathcal{H}_1$ and $\mathcal{H}_0$ is considered. Case II: if the interference background is the partial target echo embedded in noise, we have to make a decision between $\mathcal{H}_1$ and $\mathcal{H}_2$.

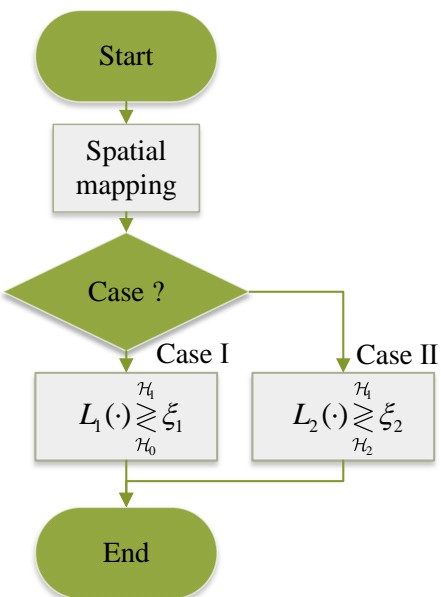

**Figure 3.** Procedure of ID-based detection method.

Case I (Decision between $\mathcal{H}_1$ and $\mathcal{H}_0$): in this case, only the decision between pure noise and a physical target is considered. For a given SRC $c(\mathbf{p}_{C,g})$, the statistic can be written as

$$
\begin{aligned}
L_1(\mathbf{Z}_g) &= \ln\left(\frac{\max\limits_{\mathbf{a}_g,\mathbf{f}_{d,g}} f(\mathbf{Z}_g|\mathbf{a}_g,\mathbf{f}_{d,g},\mathcal{H}_1)}{f(\mathbf{Z}_g|\mathcal{H}_0)}\right) \\
&= \sum_{i=0}^{I-1}\max_{\alpha_{i,g},f_{d,i,g}} -\frac{2}{\sigma_i^2}\Re\left\{\alpha_{i,g}\mathbf{d}_i^\dagger(f_{d,i,g})\mathbf{z}_{i,g}\right\} \\
&\quad + \frac{1}{\sigma_i^2}\mathbf{s}_{i,g}^\dagger\mathbf{s}_{i,g}\left|\mathbf{d}_i^\dagger(f_{d,i,g})\mathbf{d}_i(f_{d,i,g})\alpha_{i,g}\right|^2,
\end{aligned}
\tag{24}
$$

where $L_1(\mathbf{Z}_g)$ is the global statistic. The Doppler frequency is estimated autonomously by using the local signals, which dramatically reduces the complexity of the radar system. The maximum likelihood estimation (MLE) of $\alpha_{i,g}$ is given as

$$
\hat{\alpha}_{i,g} = \max_{f_{d,i,g}}\frac{\mathbf{d}_i^\dagger(f_{d,i,g})\mathbf{z}_{i,g}}{\mathbf{s}_{i,g}^\dagger\mathbf{s}_{i,g}\mathbf{d}_i^\dagger(f_{d,i,g})\mathbf{d}_i(f_{d,i,g})}.
\tag{25}
$$

Substituting (25) into (24) and simplifying, the statistic is given as

$$
L_1(\mathbf{Z}_g) = \sum_{i=0}^{I-1}\max_{f_{d,i,g}}\frac{\left|\mathbf{d}_i^\dagger(f_{d,i,g})\mathbf{z}_{i,g}\right|^2}{\sigma_i^2\mathbf{s}_{i,g}^\dagger\mathbf{s}_{i,g}\mathbf{d}_i^\dagger(f_{d,i,g})\mathbf{d}_i(f_{d,i,g})}\overset{\mathcal{H}_1}{\underset{\mathcal{H}_0}{\gtrless}}\xi_1,
\tag{26}
$$

where $\xi_1$ represents the threshold related to the false alarm rate. (26) can be considered as the original GLRT detector. However, the GLRT statistic can exceed the detection threshold under $\mathcal{H}_2$, which is defined as the additional false alarm.

Case II (Decision between $\mathcal{H}_1$ and $\mathcal{H}_2$): when $c(\mathbf{p}_{C,g}) \in \mathbb{F}(\mathbf{p}_{T,j})$, the target components from the DSRC are assumed to be the interference that makes the SRC have a high false alarm rate. The target component from the other SRC is regarded as the interference. Then, the statistic can be written as

$$L_2(\mathbf{Z}_g) = \ln\left(\frac{\max\limits_{\mathbf{a}_g, \mathbf{f}_{\mathrm{d},g}} f(\mathbf{Z}_g|\mathbf{a}_g, \mathbf{f}_{\mathrm{d},g}, \mathcal{H}_1)}{\max\limits_{\alpha_{l,g}, f_{\mathrm{d},l,g}} f(\mathbf{Z}_g|\alpha_{l,g}, f_{\mathrm{d},l,g}, \mathcal{H}_2)}\right)$$

$$= \sum_{i=0, i\neq l}^{I-1} \max_{\alpha_{i,g}, f_{\mathrm{d},i,g}} - \frac{2}{\sigma_i^2}\Re\left\{\alpha_{i,g}\mathbf{d}_i^\dagger(f_{\mathrm{d},i,g})\mathbf{z}_{i,g}\right\} \tag{27}$$

$$+ \frac{1}{\sigma_i^2}\mathbf{s}_{i,g}^\dagger\mathbf{s}_{i,g}\left|\mathbf{d}_i^\dagger(f_{\mathrm{d},i,g})\mathbf{d}_i(f_{\mathrm{d},i,g})\alpha_{i,g}\right|^2.$$

Using (20) and (25), the decision (27) rule can be rewritten as

$$L_2(\mathbf{Z}_g) = \min_l\left\{\sum_{i=0, i\neq l}^{I-1} \max_{f_{\mathrm{d},i,g}} \frac{\left|\mathbf{d}_i^\dagger(f_{\mathrm{d},i,g})\mathbf{z}_{i,g}\right|^2}{\sigma_i^2\mathbf{s}_{i,g}^\dagger\mathbf{s}_{i,g}\mathbf{d}_i^\dagger(f_{\mathrm{d},i,g})\mathbf{d}_i(f_{\mathrm{d},i,g})}\right\}\overset{\mathcal{H}_1}{\underset{\mathcal{H}_2}{\gtrless}}\xi_2, \tag{28}$$

where $\xi_2$ is the threshold in the case. For the same false alarm rate, the value of $\xi_2$ is smaller than $\xi_1$. For convinience, the decision rule (28) is defined as interference cancellation (IC)-based detector, because its statistic can be regarded as the orignal GLRT statistic minus the interferece components. Using the decision rule, we can also distinguish $\mathcal{H}_1$ from $\mathcal{H}_0$ and $\mathcal{H}_2$, because $P\{L_2(\mathbf{Z}_g) < \xi_2|\mathcal{H}_0\}$ is larger than $P\{L_2(\mathbf{Z}_g) < \xi_2|\mathcal{H}_2\}$. However, The IC-based detector has a worse target detection performance than (26), because of $P\{L_1(\mathbf{Z}_g) > \xi_1|\mathcal{H}_1\} > P\{L_2(\mathbf{Z}_g) > \xi_2|\mathcal{H}_1\}$. Thus, the final decision rule should be adjusted according to the type of background interference in order to balance the performances of target detection and addition false alarm suppression.

Given that (23), (26), and (28) all contain the term

$$\max_{f_{\mathrm{d},i,g}} \frac{\left|\mathbf{d}_i^\dagger(f_{\mathrm{d},i,g})\mathbf{z}_{i,g}\right|^2}{\mathbf{d}_i^\dagger(f_{\mathrm{d},i,g})\mathbf{d}_i(f_{\mathrm{d},i,g})}. \tag{29}$$

The solution of (29) is the indispensable. However, no analytical solutions are found for this optimization problem. Considering $\mathbf{d}_i^\dagger(f_{\mathrm{d},i,g})\mathbf{d}_i(f_{\mathrm{d},i,g}) = M$, the optimization problem that can be rewritten as

$$\max_k \frac{1}{M}|X_i(k,\kappa)|^2, \tag{30}$$

where $X_i(k,\kappa) = \sum\limits_{m=0}^{M-1} z_{i,g}[m]e^{j\frac{2\pi mk}{\kappa M}}$ denotes the discrete spectrum of the slow-time signal, which is obtained by discrete Fourier transform (DFT). $\kappa$ denotes the oversampling factor. Because $X_i(k,\kappa)$ is a sampled version of the frequency spectrum, the peak value decreases when the Doppler frequency of the target is between DFT samples, which is called a Doppler straddle loss [33]. One obvious way to reduce straddle loss is to sample the Doppler frequency axis more densely, i.e., to choose the number of spectrum samples $K > M$. The worst-case straddle loss is a function of the oversampling factor. Based on the "oversampling in Doppler frequency" method, the statistics of (23), (26), and (28) are rewritten as

$$D(\mathbf{Z}_g,\kappa) = \max_{l,k}\left\{\frac{(I-1)|X_l(k,\kappa)|^2}{\sum\limits_{i=0, i\neq l}^{I-1}|X_i(k,\kappa)|^2}\right\}, \tag{31}$$

$$L_1(\mathbf{Z}_g,\kappa) = \sum_{i=0}^{I-1}\frac{\max\limits_k|X_i(k,\kappa)|^2}{M\sigma_i^2\mathbf{s}_{i,g}^\dagger\mathbf{s}_{i,g}}, \tag{32}$$

$$L_2(\mathbf{Z}_g, \kappa) = \min_l \left\{ \sum_{i=0, i \neq l}^{I-1} \frac{\max_k |X_i(k, \kappa)|^2}{M\sigma_i^2 \mathbf{s}_{i,g}^\dagger \mathbf{s}_{i,g}} \right\}$$

$$= L_1(\mathbf{Z}_g, \kappa) - \max_{i,k} \frac{|X_i(k, \kappa)|^2}{M\sigma_i^2 \mathbf{s}_{i,g}^\dagger \mathbf{s}_{i,g}}. \tag{33}$$

The statistic (32) should be used if case I is decided and (33) is selected if case II is decided, then the detection statistic of the interference discrimination (ID)-based method is obtained as

$$L(\mathbf{Z}_g, \kappa) = \begin{cases} L_2(\mathbf{Z}_g, \kappa) & D(\mathbf{Z}_g, \kappa) > \eta \\ L_1(\mathbf{Z}_g, \kappa) & D(\mathbf{Z}_g, \kappa) < \eta \end{cases}. \tag{34}$$

Finally, ID-based detector can be viewed as a modified form of the original GLRT detector:

$$L(\mathbf{Z}_g, \kappa) = L_1(\mathbf{Z}_g, \kappa) - M(\mathbf{Z}_g, \kappa) \underset{\mathcal{H}_0 + \mathcal{H}_2}{\overset{\mathcal{H}_1}{\gtrless}} \xi, \tag{35}$$

where $M(\mathbf{Z}_g, \kappa)$ is the modified term that makes the statistic lower than the detection threshold under $\mathcal{H}_2$. The modified term can be considered an estimate of the interference component, which can be written as

$$M(\mathbf{Z}_g, \kappa) = \begin{cases} \max_{i,k} \dfrac{|X_i(k, \kappa)|^2}{M\sigma_i^2 \mathbf{s}_{i,g}^\dagger \mathbf{s}_{i,g}} & D(\mathbf{Z}_g, \kappa) > \eta \\ 0 & D(\mathbf{Z}_g, \kappa) < \eta \end{cases}. \tag{36}$$

The term $\xi$ denotes the detection threshold. The test statistics might change after discrimination, and the stationary threshold cannot be applied to the modified statistics. Therefore, the detection threshold should be adjusted according to the discrimination result.

*3.2. Performance Analysis*

In this section, we provide a statistical analysis of the ID-based detector and the original GLRT detector in condition of $\alpha_{i,j} \sim \mathcal{CN}(0, \sigma_{\mathrm{T},j}{}^2)$ and $\kappa = 1$. The parameter settings are standard in the analysis of radar detection when the spatial diversity is exploited and the straddle loss is negligible. The value of $\kappa$ dominates computational complexity, so the balance between the detection performance and the computational cost should be considered in practice. Actually, we need the PDF of $K$ correlated Gaussian random variables to analyze the detection performance, because the PDF is difficult to determine for different values of $\kappa$ [43]. The analysis can provide a benchmark of a achievable detection performance.

3.2.1. Performance of Original Detector

The statistic of the original GLRT detector is given by (32) where only the null hypothesis and non-null hypothesis are considered. Our main results are included by the following theorem. Given the signal model in (12), the PDFs of the statistic under all hypotheses are given respectively by

$$f_{L_1}(x|\mathcal{H}_0) = \int_{-\infty}^{+\infty} \left[ \sum_{m=0}^{M-1} \binom{m}{M-1} \frac{M}{m+1+jw} \right]^I e^{jwx} \mathrm{d}w, \tag{37}$$

$$f_{L_1}(x|\mathcal{H}_1) = \int_{-\infty}^{+\infty} \prod_{i=0}^{I-1} \sum_{m=0}^{M-1} \binom{m}{M-1} \frac{M}{m+1+jw(1+\rho_{i,j})} e^{jwx} \mathrm{d}w, \tag{38}$$

$$\begin{aligned} f_{L_1}(x|\mathcal{H}_2) = \int_{-\infty}^{+\infty} &\left[ \sum_{m=0}^{M-1} \binom{m}{M-1} \frac{M}{m+1+jw} \right]^{I-1} \\ &\times \sum_{m=0}^{M-1} \binom{m}{M-1} \frac{M}{m+1+jw(1+\rho_{i,l})} e^{jwx} \mathrm{d}w, \end{aligned} \tag{39}$$

with several derivations (also see in Appendix A). $j = \sqrt{-1}$. $\rho_{i,j} = \sigma_{\mathrm{T},j}{}^2/\sigma_i{}^2$ represents the signal to noise ratio (SNR).

The right-tail probability can be obtained by $Q_0(\xi, I) = 1 - \int_0^\xi f_{L_1}(x|\mathcal{H}_0)\mathrm{d}x$, $Q_1(\xi, I) = 1 - \int_0^\xi f_{L_1}(x|\mathcal{H}_1)\mathrm{d}x$, and $Q_2(\xi, I) = 1 - \int_0^\xi f_{L_1}(x|\mathcal{H}_0)\mathrm{d}x$. $Q_0(\xi, I)$ and $Q_1(\xi, I)$ are also referred to the probability of false alarm and target detection, respectively. $Q_2(\xi, I)$ represents the probability of additional false alarm, i.e., the probability of mistaking "ghost target" for a physical target.

### 3.2.2. Performance of ID-Based Detector

The statistic of the proposed method is given in (35). We provide the analysis about the probability of false alarm, target detection and the additional false alarm.

The proposed method discriminates the type of background interference before a final decision. As a result, the performance is related to the performance of the background discrimination. The performance of the discriminator is provided by Monte Carlo (MC) simulations in Section 4. First, the target detection probability is provided:

$$\begin{aligned} P_{\mathrm{d}} &= P\{\mathcal{H}_1|\mathcal{H}_1\} \\ &= P\{\mathcal{H}_1|\text{case I}, \mathcal{H}_1\} P\{\text{case I}|\text{case I}\} \\ &\quad + P\{\mathcal{H}_1|\text{case II}, \mathcal{H}_1\} P\{\text{case II}|\text{case I}\} \\ &= (1-P_{\mathrm{e}})Q_1(\xi, I) + P_{\mathrm{e}}Q_1'(\xi, I), \end{aligned} \tag{40}$$

where $Q_1'(\xi, I)$ represents the right-tail probability of (27) under $\mathcal{H}_1$. If the $P_{\mathrm{e}}$ is negligible, we have $P_{\mathrm{e}}Q_1'(\xi, I) << (1-P_{\mathrm{e}})Q_1(\xi, I)$. The probability of target detection can be simplified approximately:

$$P_{\mathrm{d}} \approx (1-P_{\mathrm{e}})Q_1(\xi, I). \tag{41}$$

From (41), we note that the ID based-detector has a slight performance of target detection. Larger $P_{\mathrm{e}}$ will bring in more severe target detection performance loss. Similarly, the probability of false alarm can be obtained by

$$\begin{aligned} P_{\mathrm{fa}} &= P\{\mathcal{H}_1|\text{case I}, \mathcal{H}_0\} P\{\text{case I}|\text{case I}\} \\ &\quad + P\{\mathcal{H}_1|\text{case II}, \mathcal{H}_0\} P\{\text{case II}|\text{case I}\} \\ &\approx (1-P_{\mathrm{e}})Q_0(\xi, I). \end{aligned} \tag{42}$$

The false alarm rate reduces slightly compared with the original GLRT detector. In practice, the false alarm keeps a very low level, so we have $P_{\mathrm{e}}Q_0(\xi, I) << Q_0(\xi, I)$. The false alarm rate of the two detectors are approximately equal, i.e., $P_{\mathrm{fa}} \approx Q_1(\xi, I)$.

Finally, the probability of additional false alarm is given by

$$\begin{aligned} P_{\mathrm{afa}} &= P\{\mathcal{H}_1|\text{case I}, \mathcal{H}_2\} P\{\text{case I}|\text{case II}\} \\ &\quad + P\{\mathcal{H}_1|\text{case II}, \mathcal{H}_2\} P\{\text{case II}|\text{case II}\} \\ &= (1-P_{\mathrm{c}})Q_2(\xi, I) + P_{\mathrm{c}}Q_2'(\xi, I), \end{aligned} \tag{43}$$

where $P_c = P\{\text{case II}|\text{case II}\}$ denotes the probability of correct discrimination, which is the function of SNR. $Q_2'(\xi)$ represents the complementary cumulative distribution function of (27) under $\mathcal{H}_2$. We only consider the statistical analysis of extreme case in this subsection, and the details of performance analysis are provided based on the numerical in Section 4. When the SNR is high enough to estimate the interference amplitude, we have $P_{\text{afa}} \approx Q_0(\xi, I - 1)$. When the SNR is low enough, the additional false alarm rate can be obtained by $P_{\text{afa}} \approx Q_2(\xi, I - 1)$.

## 4. Numerical Experiment

In this section, numerical simulation results are presented to verify the above analysis. First, we simulate power superposition based on the space mapping scheme and analyze the "ghost target" problem. Then, the performance analysis of background discrimination is provided. Finally, the detection results of ID-based detector are presented compared with the original GLRT detector.

The scenario where three targets and an MSRS exist is considered to simulate spatial power accumulation. The deployment of the MSRS, which involves seven radars, is illustrated in Figure 4. These radar sites are located at $(4.4, -30)$, $(1.8, -20)$, $(0.8, -10)$, $(0, 0)$, $(0.8, 10)$, $(1.8, 20)$, and $(4.4, 30)$ km. The three targets are placed at $(49, 1.2)$, $(50, 2.0)$, and $(52, 1.6)$ km and move with velocities of $(-122, -111)$, $(-110, -92)$, and $(-100, -106)$ m/s, respectively. All radar work with the same parameters. The beam direction is specially allocated so that all targets can be observed. The signal bandwidth is 5 MHz, and the duration is 51 μs. The transmitted signals are satisfy $\mathbf{s}_i^\dagger \mathbf{s}_i = 1$. The pulse number is 16, the PRI is 1 ms, and the beamwidth is 4 deg. The amplitudes of target follows the complex Gaussian distribution with zero mean and $\sigma_{\text{T},j}^2$ variance.

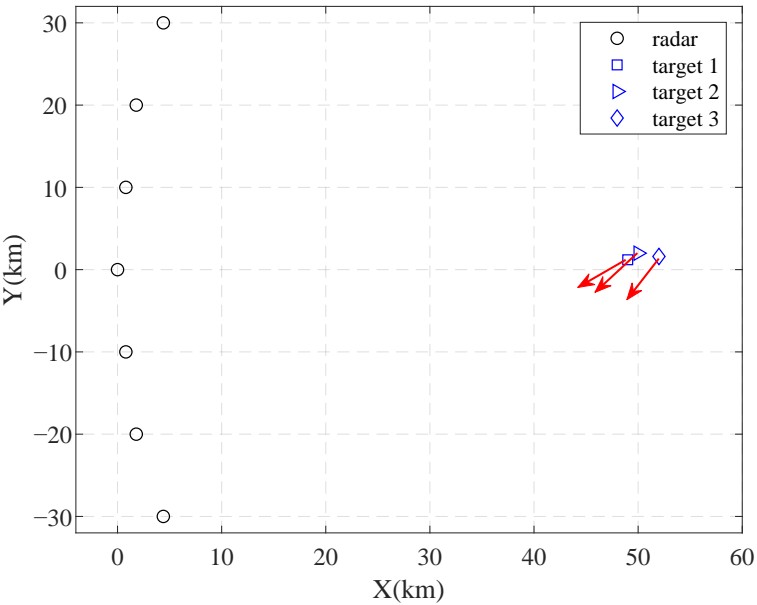

**Figure 4.** Placement of the MSRS and targets.

### 4.1. Power Superposition and "Ghost Target"

The peak values of the power superposition are located at the SRCs containing targets, while partial power of the targets is gathered in the RSRCs subject to the DSRCs. The incomplete power accumulation is caused by the data associate error and makes the statistic exceed the detection threshold with high probability. That is why the "ghost target" problem occurs unexpectedly.

Figure 5 illustrates the spatial power superposition when the SNR value is set to 2 dB. The statistic (24) can be considered power superposition method. Taking the target at $(50, 2.0)$ km as an example, the spatial power superposition is approximately 22.8 dB. To

analyze the symbiosis effect, we typically take the SRC at (50.93, 1.95) km as an example. The SRC at (50.93, 1.95) km is the RSRC corresponding to target 3. The power accumulation is 15.1 dB, which is higher than the noise level. If we decide based on orignal GLRT detector, the RSRC may be treated as a physical target, which is a "ghost target".

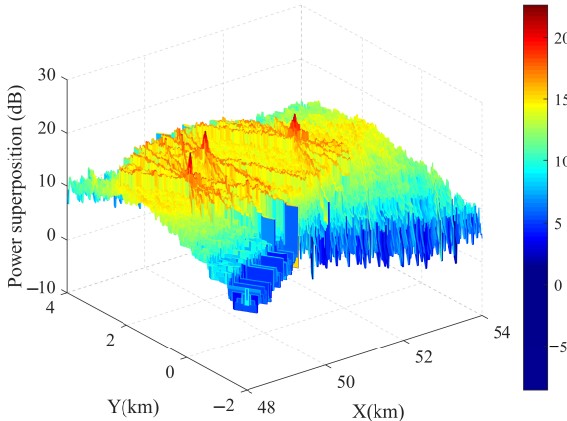

**Figure 5.** Energy accumulation within common surveillance region and $\kappa = 4$ and SNR = 2 dB.

The false alarm rate is given for different oversampling factors $\kappa$ in Figure 6. The threshold can be obtained by the MC results. In fact, a smaller search step is taken when $\kappa$ becomes larger. Thus, the false alarms of the detector increase with the reduction of the search step.

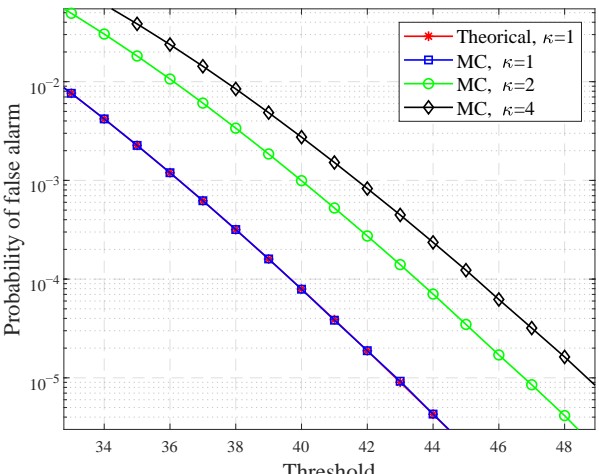

**Figure 6.** $P_{\text{fa}}$ versus the test threshold.

Figure 7 depicts the detection performance of the original GLRT detector under the condition of $P_{\text{fa}} = 10^{-5}$. We can provide the best benchmark of target detection through analyzing the result. Although the threshold will be enhanced in the condition of oversampling, the detector has better detection performance. Consequently, "oversampling" could improve the detection performances of the detector with a limit.

Figure 8 depicts the detection probability of the SRC at (50.93, 1.95) km for different oversampling factors $\kappa$. The detection probability of the RSRC describes the possibility of a "ghost target", i.e., the probability of additional false alarm. Similarly, the probability rises as $\kappa$ increases. Although DSRC can be detected more easily than its RSRCs, the RSRCs can also be detected with a certain probability that should not be ignored. The higher the probability of RSRC detection is, the more "ghost targets" can be detected. Because of the "ghost target", the number and physical position of targets can hardly be obtained after all observation data are tested.

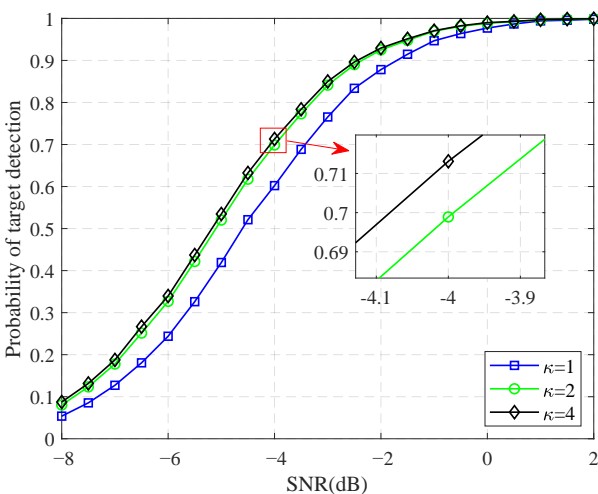

**Figure 7.** Detection probability when $P_{\text{fa}} = 10^{-5}$.

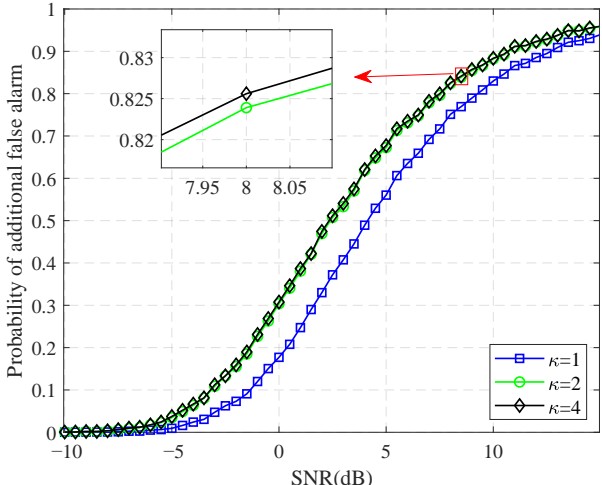

**Figure 8.** Detection probability of the SRC at (50.93, 1.95) km when $P_{\text{fa}} = 10^{-5}$.

### 4.2. Background Interference Discrimination

As previously mentioned, the discrimination of background interference is the prerequisite of the proposed algorithm. The discrimination performance is crucial to the final decision of the ID-based detector. Therefore, we analyze the performance of the discriminator before performing the final detection.

The threshold of the discriminator is designed on the basis of the principle of constant misjudgment probability, i.e., constant $P_{\text{e}}$. Figure 9 shows that the thresholds obtained using MC approach. The probability of a correct decision $P_{\text{c}} = P\{\text{case II}|\text{case II}\}$ is used to evaluate the performance of the discriminator. Figure 10 depicts the probability of the correct discrimination under different misjudgment probabilities and oversampling factor. The performance of the discriminator deteriorates as the SNR decreases. In addition, a smaller misjudgment probability signifies a larger discrimination threshold. Therefore, the probability of achieving a correct decision is high if the misjudgment probability is large.

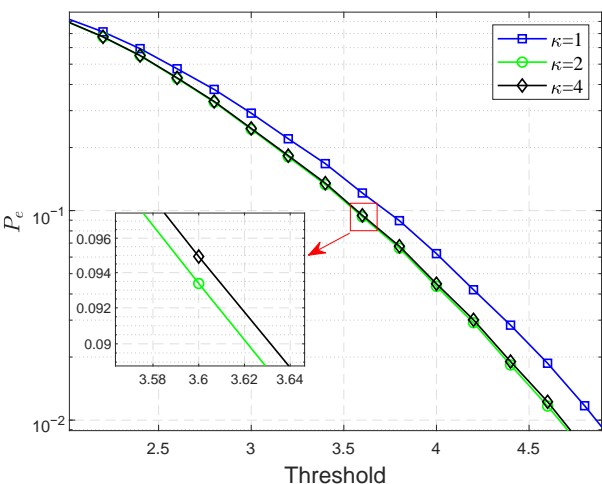

**Figure 9.** $P_e$ versus discrimination threshold.

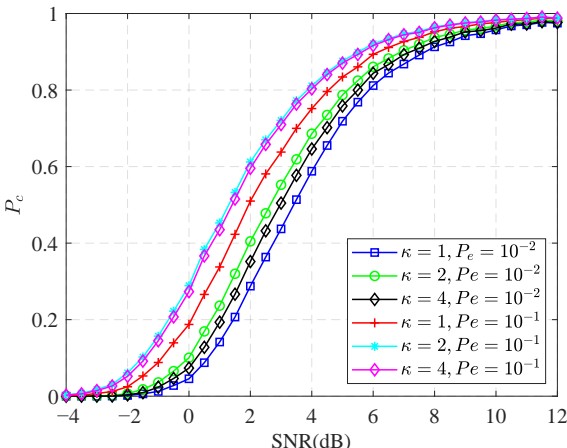

**Figure 10.** $P_e$ Probability of correct decision in the interference discrimination.

### 4.3. Comparison of Original GLRT and ID-Based Detector

The total processes of ID-based detection method are simulated under the condition of SNR = 2 dB and $P_{fa} = 10^{-5}$. The ID-based detection method is compared with the original GLRT detector and IC-based detector. Figures 11 and 12 depict the detection results with three targets and five targets respectively. The experiment in Figure 12 employs two extra targets which are located at (51, 1.3) and (48.5, 0.8) km with the velocities (−120, 100) and (−95, −95) m/s. The bright spot represents the plot detected by the radar system. The centers of the red cycles are the physical targets, whereas the other spots represent false alarms. Figures 11a and 12a show that the original GLRT detector generates "ghost targets" because the statistics of the RSRCs can also exceed the detection threshold with a certain probability. The "ghost target" problem inhibits estimating the number and positions of targets after detection. In comparison, the ID-based and IC-based detectors detect all targets correctly with few "ghost targets", as shown in Figures 11b,c, and 12b,c.

For the multitarget detection problem, we should analyze not only target detection but also suppression of the "ghost target". Therefore, the probability of target detection and additional false alarm are taken to describe the performance of the detection method. Figure 13 illustrates the probability of target detection of the original GLRT detector IC-based detector,and ID-based detector. All targets are detected dependently. The IC algorithm has a big performance in target detection because of partial target components may be cancelled. The detection performance of ID slightly declines as $P_e$ increases. A larger $P_e$ means that targets are considered RSRCs more easily, which leads to energy loss.

As shown in Figure 14, the probability of additional false alarm for the original and ID-based detector are simulated at different $P_e$ when the false alarm rate is set to $10^{-6}$, $10^{-5}$, and $10^{-4}$. Compared with the IC-based method, the ID-based detector is performed better at target detection and worse at supressing "ghost targets". The higher the SNR is, the more "ghost targets" are generated in the original GLRT detector. In contrast, the proposed method maintains a low probability when the SNR is low enough to be treated as noise and high enough to discriminate the RSRC corresponding to a target. Figure 14a shows the probability of additional false alarm. The highest probability occurs at SNR = 4 dB, where the physical target is probably treated as the "ghost target". This performance degradation becomes severe in Figure 14b,c. As $P_{fa}$ increases, the signal gathered in the RSRC with a low SNR can be detected. If the SNR is not sufficiently large to distinguish the "ghost target" from the noise background, the RSRC will be detected with a high probability. Therefore, the "ghost target" suppression performance is influenced by $P_e$ and $P_{fa}$ in the ID-based detection method. The $P_e$ that matches $P_{fa}$ should be selected to obtain satisfactory additional false alarm and target performance in the real world.

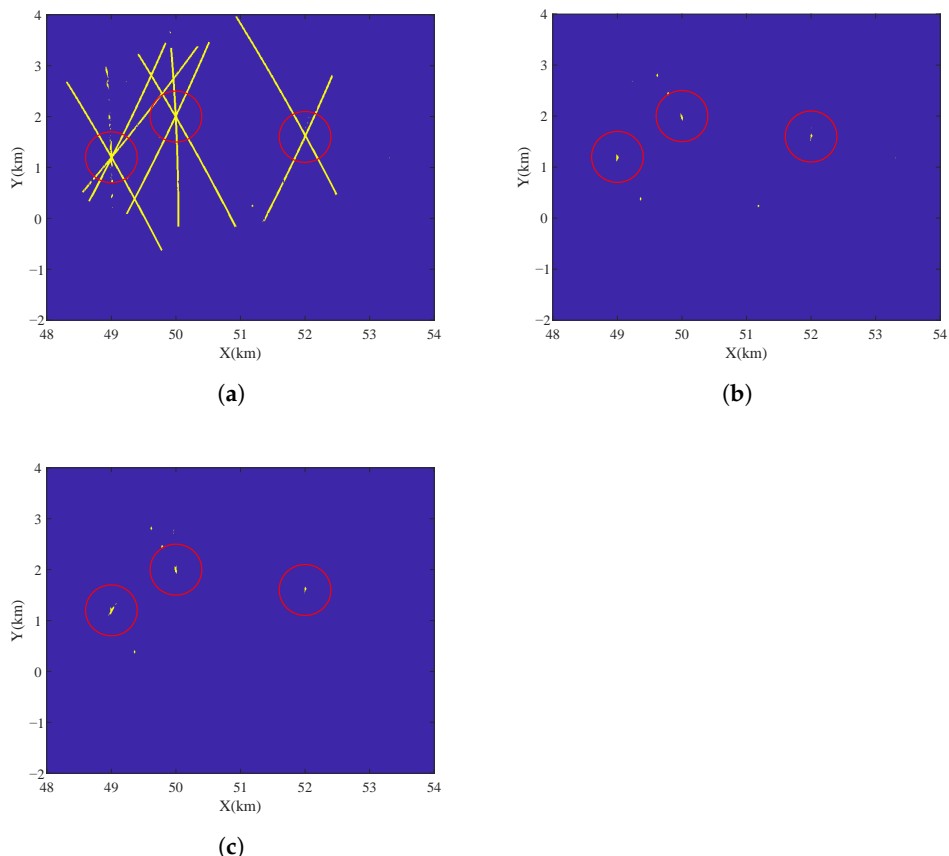

(a)

(b)

(c)

**Figure 11.** Detection result with three targets when SNR = 2dB. (**a**) The original GLRT detector. (**b**) ID-based detector. (**c**) IC-based detector.

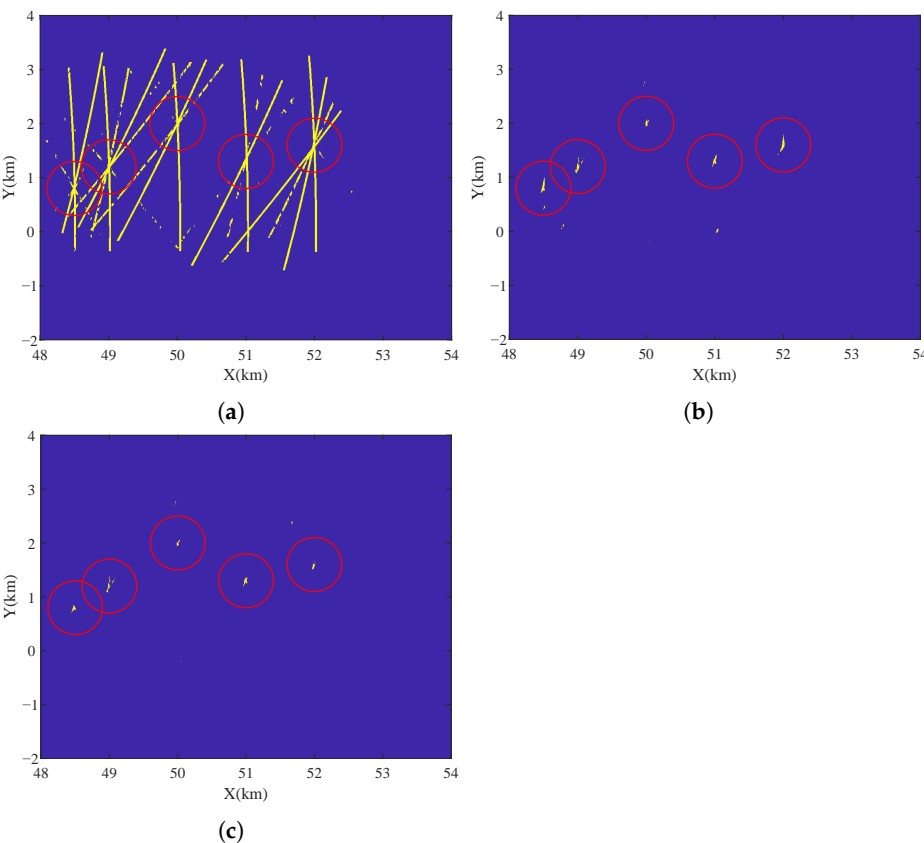

**Figure 12.** Detection result with five targets when SNR = 2dB. (**a**) The original GLRT detector. (**b**) ID-based detector. (**c**) IC-based detector.

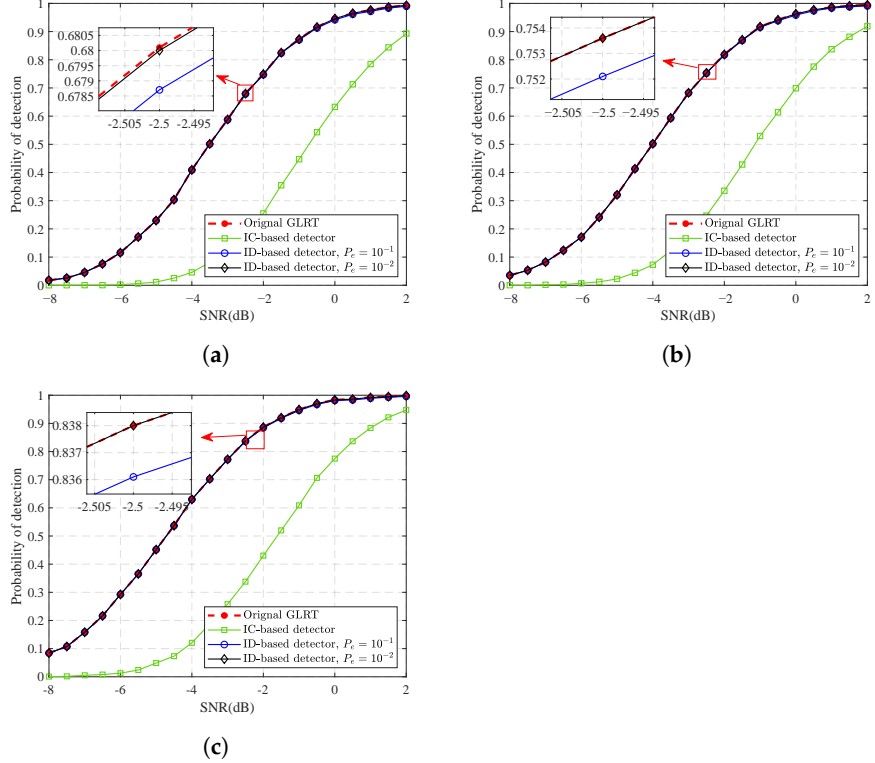

**Figure 13.** Probability of target detection for different $P_e$ when $\kappa = 1$. (**a**) $P_{\text{fa}} = 10^{-6}$. (**b**) $P_{\text{fa}} = 10^{-5}$. (**c**) $P_{\text{fa}} = 10^{-4}$.

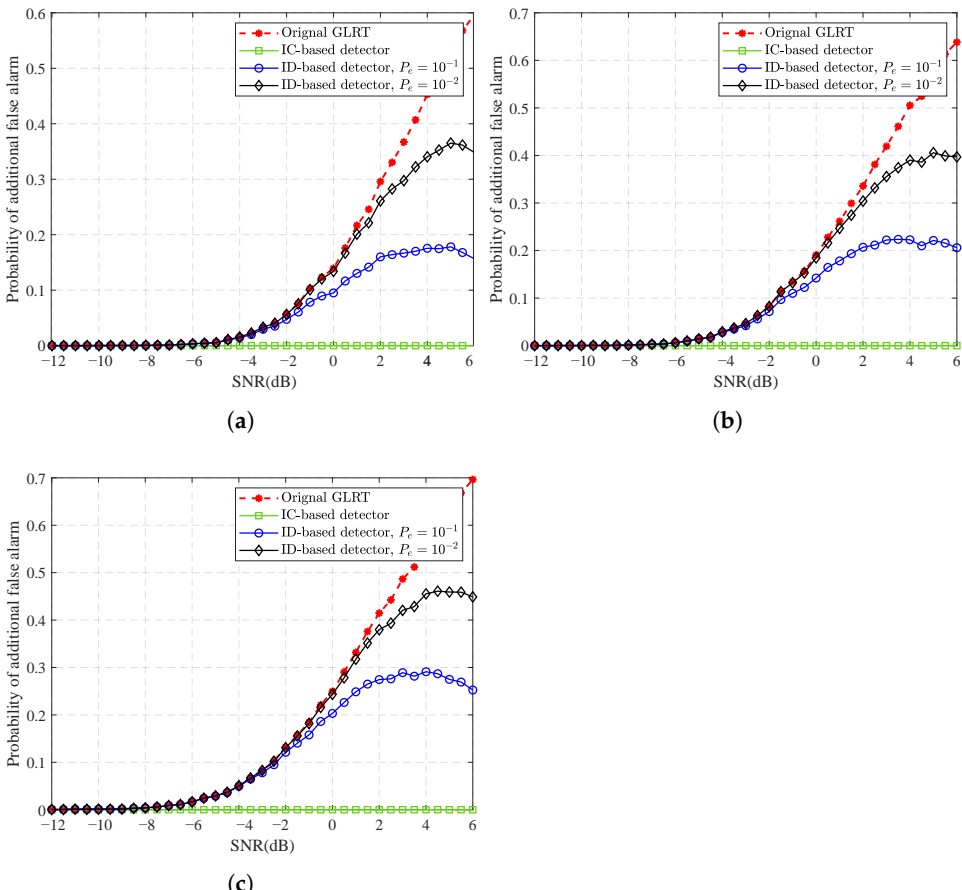

**Figure 14.** Probability of additional false alarm $\kappa = 1$. (**a**) $P_{\text{fa}} = 10^{-6}$. (**b**) $P_{\text{fa}} = 10^{-5}$. (**c**) $P_{\text{fa}} = 10^{-4}$.

## 5. Conclusions

In this study, we investigate moving target detection of multistatic radar for multiple targets in the common surveillance region. To combat the additional false alarm caused by spatial mapping, the detection problem is built as the ternary hypothesis testing where the physical target is distinguished from "ghost target" and pure noise. An ID-based detection method is proposed, where background interference is first discriminated and final decision rule is adjusted accordingly. Theoretical analysis has shown that the ID-based detector and the GLRT detector have approximate performances of target detection and false alarm, but the ID-based detector can suppress the additional false alarm. Simulation results verify our analysis and show that proposed method can detect multiple moving targets while suppressing the "ghost target" when $P_{\text{e}}$ that matches $P_{\text{fa}}$ is selected. Because an SRC is assumed to contain an individual target in our work. The performances of our proposed method may decline, when targets exist in the same SRC and are unresolved by MSRS. In the future, the problem will be solved and a real-life experiment will be performed.

**Author Contributions:** S.Z. and Y.Z. were involved in the computational framework, conceptualization, methodology, data analysis, results interpretation, and paper writing; M.S. simulated the experiments and processed the measured data; L.D. and L.Z. were involved in methodology; L.Z. had the general supervision of the study and provided solutions to any problems that arose. All authors have read and agreed to the published version of the manuscript.

**Funding:** This work was funded in part by the National Natural Science Foundation of China under Grant 61871305, Grant 62171336, and Grant 61731023.

**Data Availability Statement:** Not applicable.

**Acknowledgments:** The authors would like to thank the anonymous Reviewers for their valuable comments, which improved the paper's quality.

**Conflicts of Interest:** The authors declare no conflict of interest.

## Appendix A

In the appendix, we list the derivation of (37)–(39). Under $\mathcal{H}_0$, the random variables $X_i(k, 1)$ for $k = 0, 1, \ldots, K - 1$ are dependent and Gaussian [43]. They have the mean zero and the variance $M\sigma_i^2 \mathbf{s}_{i,g}^\dagger \mathbf{s}_{i,g}$. Thus,

$$l_i\left(\mathbf{z}_{i,g}, 1\right) = \frac{|X_i(k, 1)|^2}{M\sigma_i^2 \mathbf{s}_{i,g}^\dagger \mathbf{s}_{i,g}} \sim \chi_2^2, \tag{A1}$$

where $\chi_2^2$ denotes the chi-squared distribution. The PDF of $l_i\left(\mathbf{z}_{i,g}, 1\right)$ can be obtained as

$$\begin{aligned} f_{l_i}(x|\mathcal{H}_0) &= \frac{\mathrm{d}Q_{\chi_2^2}^{K-1}(x)}{\mathrm{d}x} \\ &= M\exp(-x)[1 - \exp(-x)]^{M-1}, \end{aligned} \tag{A2}$$

with $x > 0$. Note that the global statistic of DML-GLRT is the sum of the local statistics. Using the convolution theorem, the PDF of $L_1(\mathbf{Z}_g, 1)$ can be given as

$$\begin{aligned} f_{L_1}(x|\mathcal{H}_0) &= \overset{I-1}{\underset{i=0}{\otimes}} f_{l_i}(x|\mathcal{H}_0) \\ &= \mathcal{F}^{-1}\left\{ \mathcal{F}\left\{ \prod_{i=0}^{I-1} f_{l_i}(x|\mathcal{H}_0) \right\} \right\} \\ &= \int_{-\infty}^{+\infty} \left[ \sum_{m=0}^{M-1} \binom{m}{M-1} \frac{M}{m+1+jw} \right]^I e^{jwx} \mathrm{d}w. \end{aligned} \tag{A3}$$

Assume that $\alpha_{i,j}$ is a dependent Gaussian variable with zero mean and variance $\sigma_{\mathrm{T},j}^2$. $\mathcal{F}\{\cdot\}$ and $\mathcal{F}^{-1}\{\cdot\}$ denotes the Fourier transform and the inverse Fourier transform, respectively. Similarly, the conditional PDF of $l_i\left(\mathbf{z}_{i,g}, 1\right)$ under $\mathcal{H}_1$ is expressed as

$$f_{l_i}(x|\mathcal{H}_1) = M\exp\left(-\frac{x}{\rho_{i,j}}\right)\left[1 - \exp\left(-\frac{x}{\rho_{i,j}}\right)\right]^{M-1}, \tag{A4}$$

where $\rho_{i,j} = 1 + \sigma_{\mathrm{T},j}^2 / \sigma_i^2$.

Therefore, the global statistic under $\mathcal{H}_1$ is obtained as

$$\begin{aligned} f_{L_1}(x|\mathcal{H}_1) &= \overset{I-1}{\underset{i=0}{\otimes}} f_{l_i}(x|\mathcal{H}_1) \\ &= \int_{-\infty}^{+\infty} \prod_{i=0}^{I-1} \sum_{m=0}^{M-1} \binom{m}{M-1} \frac{M}{m+1+jw(1+\rho_{i,j})} e^{jwx} \mathrm{d}w. \end{aligned} \tag{A5}$$

Under $\mathcal{H}_2$, the PDF of $l_i\left(\mathbf{z}_{i,g}, 1\right)$ is

$$f_{l_i}(x|\mathcal{H}_2) = \begin{cases} M\exp\left(-\dfrac{x}{\rho_{i,j}}\right)\left[1 - \exp\left(-\dfrac{x}{\rho_{i,j}}\right)\right]^{M-1} & i = l \\ M\exp(-x)[1 - \exp(-x)]^{M-1} & i \neq l, \end{cases} \tag{A6}$$

The PDF of the global statistic is

$$
\begin{aligned}
f_{L_1}(x|\mathcal{H}_2) &= \overset{I-1}{\underset{i=0}{\otimes}} f_{l_i}(x|\mathcal{H}_2) \\
&= \int_{-\infty}^{+\infty} \left[ \sum_{m=0}^{M-1} \binom{m}{M-1} \frac{M}{m+1+jw} \right]^{I-1} \\
&\quad \times \sum_{m=0}^{M-1} \binom{m}{M-1} \frac{M}{m+1+jw(1+\rho_{i,l})} e^{jwx} \mathrm{d}w.
\end{aligned}
\tag{A7}
$$

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
