# Peer review of "Moving Multitarget Detection Using a Multisite Radar System with Widely Separated Stations"

_remotesensing, doi:10.3390/rs14112660_

Round 1

Reviewer 1 Report

In this contribution the detection problem has been investigated. My opinion is that the contribution has merit, and the related findings can be very useful to the scientific community. Also the overall organization and English language are good.
I only suggest to perform a critical analysis of the already published literature and highlight the novelty of the contribution.
Moreover, the authors might better describe the radar operating mode, that has not been clarified. To this aim, I suggest the following reference, when some theoretical concepts are explained.

E. Cardillo, C. Li and A. Caddemi, "Millimeter-Wave Radar Cane: A Blind People Aid With Moving Human Recognition Capabilities," in IEEE Journal of Electromagnetics, RF and Microwaves in Medicine and Biology, doi: 10.1109/JERM.2021.3117129.

Author Response

 We would like to thank you for the excellent comments and for taking
the time to consider our paper. Those comments are all valuable and very helpful for revising and improving our paper. We have studied the comments carefully and tried our best to revise the manuscript.  The detailed point-by-point responses are provided in the attachment. Please see the attachment.

Reviewer 2 Report

This paper proposes a spatial mapping to integrate the observation data for the detection problem of multiple moving targets using a multisite radar system with widely separated stations. Simulated analysis is presented to verify the effectiveness of the proposed method.

Overall, the topic of this paper is interesting, and the study is both complete and convincing. I have the following concerns:

  1. Abstract: It is well organized and clearly presented. It is a bit long for the problem description, it’s better to be condensed. In addition, the major contribution of this work should be highlighted here.

  1. Introduction: I think many recent new detectors can be added into this part to enrich the content, such as [1-3].

[1] X. Hua, Y. Ono, L. Peng, Y. Cheng and H. Wang, "Target Detection Within Nonhomogeneous Clutter Via Total Bregman Divergence-Based Matrix Information Geometry Detectors," in IEEE Transactions on Signal Processing, vol. 69, pp. 4326-4340, 2021, doi: 10.1109/TSP.2021.3095725.

[2] J. Liu, W. Liu, Y. Gao, S. Zhou and X. -G. Xia, "Persymmetric Adaptive Detection of Subspace Signals: Algorithms and Performance Analysis," in IEEE Transactions on Signal Processing, vol. 66, no. 23, pp. 6124-6136, 1 Dec.1, 2018, doi: 10.1109/TSP.2018.2875416.

[3] K. Ghojavand, M. Derakhtian and M. Biguesh, "Rao-Based Detectors for Adaptive Target Detection in the Presence of Signal-Dependent Interference," in IEEE Transactions on Signal Processing, vol. 68, pp. 1662-1672, 2020, doi: 10.1109/TSP.2020.2969047.

  1. Please cite some references for the existence equations.

  1. It’s better to give more comparison results on simulation and real dataset about Pd vs SNRs to improve the convincingness of your proposed methods.

5. Conclusions: The authors should point out the potential limitations of their work and further improvements.

Author Response

(The authors gave the same response as above.)

Reviewer 3 Report

Moving Multitarget Detection Using a Multisite Radar System

with Widely Separated Stations

This manuscript described a spatial mapping to integrate the observation data of a moving target from different angles into a spatial resolution cell. And the author said that the proposed method, taking the advantages of the ternary hypothesis testing could achieve accurately multiple moving targets detection while suppressing the ghost target.

Here are some suggestions for the authors:

  1. In Section 1, the main contributions of this paper should be list.
  2. Applicability of the algorithm should be stated. The number of target detection takes 3 as an example, which makes the universality of the algorithm questioned. The amount of data is slightly too small to validate the effectiveness of the proposed method.
  3. Comparative experiments should be added.
  4. Finally, an extensive English review is required. I suggest the author to seek assistance from the colleague whose native language is English or any other professional copy-editing service to improve the quality of the paper. 

In short, this manuscript seems good idea but shows lack of comparative experiments results, and the description of algorithm robustness. Therefore, it brings me to this conclusion that the manuscript cannot be proceeded for publishing in Remote Sensing.

Author Response

(The authors gave the same response as above.)

Round 2

Reviewer 2 Report

The authors have addressed all my concerns.

Reviewer 3 Report

The author has answered all my questions. Well done.